# A fly model establishes distinct mechanisms for synthetic CRISPR/Cas9 sex distorters

**Barbara Fasulo**[1], **Angela Meccariello**[1], **Maya Morgan**[1], **Carl Borufka**[1], **Philippos Aris Papathanos**[2]*, **Nikolai Windbichler**[1]*

**1** Department of Life Sciences, Imperial College London, Sir Alexander Fleming Building, South Kensington Campus, London, United Kingdom, **2** Department of Entomology, Robert H. Smith Faculty of Agriculture, Food and Environment, Hebrew University of Jerusalem, Rehovot, Israel

* p.papathanos@mail.huji.ac.il (PAP); nikolai.windbichler@imperial.ac.uk (NW)

**Data Availability Statement:** All relevant data are within the manuscript and its Supporting Information files.

## Abstract

Synthetic sex distorters have recently been developed in the malaria mosquito, relying on endonucleases that target the X-chromosome during spermatogenesis. Although inspired by naturally-occurring traits, it has remained unclear how they function and, given their potential for genetic control, how portable this strategy is across species. We established *Drosophila* models for two distinct mechanisms for CRISPR/Cas9 sex-ratio distortion—"X-shredding" and "X-poisoning"—and dissected their target-site requirements and repair dynamics. X-shredding resulted in sex distortion when Cas9 endonuclease activity occurred during the meiotic stages of spermatogenesis but not when Cas9 was expressed from the stem cell stages onwards. Our results suggest that X-shredding is counteracted by the NHEJ DNA repair pathway and can operate on a single repeat cluster of non-essential sequences, although the targeting of a number of such repeats had no effect on the sex ratio. X-poisoning by contrast, i.e. targeting putative haplolethal genes on the X chromosome, induced a high bias towards males (>92%) when we directed Cas9 cleavage to the X-linked ribosomal target gene *RpS6*. In the case of X-poisoning sex distortion was coupled to a loss in reproductive output, although a dominant-negative effect appeared to drive the mechanism of female lethality. These model systems will guide the study and the application of sex distorters to medically or agriculturally important insect target species.

## Author summary

Harmful insect populations can be eliminated for a lack of females if they are made to produce mostly male offspring. There are genes that occur naturally that make males produce mostly sons and, although we don't know exactly how they work, this appears to coincide with damage to the X-chromosome during the production of sperm. Recently, we showed in a mosquito species that such sex-biasing genes could also be constructed artificially from first principles. To better understand if this works in other species too, we designed and built male-biasing genes of two types in the fruit fly and determined what is needed to for a shift towards males. We show how different ways of cutting the X-chromosome DNA at different times with CRISPR, results in distinct outcomes and started to ask what

**Funding:** This study was funded by the BBSRC under the research grant BB/P000843/1 to NW. PAP was funded by the Italian Ministry Education, University and Research (MIUR—D.M. no. 79 04.02.2014), by the United States – Israel Binational Agricultural Research and Development Fund (Research Grant No. IS-5180-19) and by the Israel Science Foundation (Research Grant No. 2388/19). The funders had no role in study design, data collection and analysis, decision to publish, or preparation of the manuscript.

**Competing interests:** The authors have declared that no competing interests exist.

cellular processes are involved in this. These models will help us to design such genes for the control of insect species that transmit disease or threaten crops.

## Introduction

In a population of sexually reproducing organisms, a significant sex bias towards males is predicted to decrease the population's overall reproductive output. One factor is the fecundity of females, the sex with a lower rate of gamete production, which can exert a large influence on the size of a population. Forcing the sex-ratio towards males has thus long been regarded as a potential avenue for the genetic control of harmful insect pest or disease-vector populations. While the introduction of various forms of female-killing genetic traits could achieve this goal, more powerful strategies have also been theorized. Hamilton, for example, speculated that a population of a heterogametic species would become increasingly male-biased if, at each generation a mutant Y chromosome would favour its transmission over the X-chromosome. The decline in female numbers would result in a reduced population size and eventually the collapse of the population [1]. Hamilton's thinking was inspired by records of distorter traits in *Aedes aegypti* and *Culex pipiens* that can produce extreme sex-ratios of >90% males. Cytological observations during male meiosis showed broken X-chromosomes suggesting a causal link with the male-bias phenotype [2, 3]. These findings inspired the generation of artificial distorter traits [4] first by using His-Cys box homing endonucleases and subsequently RNA-guided endonucleases. In the malaria vector *Anopheles gambiae* (*A. gambiae*) autosomal I-PpoI [5] or CRISPR/Cas9-bearing transgenes [6] were used to target sequences on the X-chromosome during male meiosis. Even if these autosomal X-shredders were self-limiting and thus less invasive and powerful compared to a driving Y chromosome, cage experiments with I-PpoI induced extreme male-biased sex-ratios and population collapse confirming the potential of this system for genetic control.

Mechanistically, the nature of the target locus for which successful X-shredding was demonstrated suggested a possible coalescence of different effects in the mosquito system. The target sites are situated within the *A. gambiae* 28S rDNA cluster which simultaneously represents (i) a high-copy number repeat on the X-chromosome, (ii) an essential gene for ribosome biogenesis and function, (iii) the nucleolar organizing region of the cell as well as (iv) a sequence adjacent to the centromere of the X-chromosome and (v) the predicted pseudo-autosomal region of the X-chromosome mediating pairing with the Y chromosome during meiosis [7]. The possible conflation of effects induced by X-shredding in the mosquito has been a source of uncertainty regarding the potential to transfer this paradigm to other important pest species. In particular, the X-shredding approach has not been clearly delineated from a related, recently-proposed strategy, based on the targeting of X-linked haploinsufficient genes (commonly ribosomal genes) with the intent to induce female lethality [8]. To delineate this mechanism, which is assumed not to alter gamete production and to come into effect only in the developing progeny, we refer to it as X-poisoning. X-poisoning would also generate a male biased progeny but would be expected to lead to a significant loss of reproductive output in the form of inviable female embryos. Reduced hatching is however a feature of the mosquito sex distortion system in some transgenic strains [9]. Although experiments suggested that carry over of endonuclease protein rather than insufficiency of the rDNA was responsible for this zygotic lethality [4], a contribution of altered target gene function to this effect could not be ruled out completely.

Here, we have sought to disentangle and reconstitute these two strategies in *Drosophila melanogaster* by targeting with CRISPR/Cas9 both X-linked multicopy repeats and, in parallel, X-linked putative haplolethal genes essential for ribosome function. By doing so we have also sought to demonstrate that the X-shredding mechanism is transferable, in principle, between species and between target genes and has potential applications beyond malaria vector control. We have also explored the efficiency of X-poisoning in biasing the sex-ratio as an alternative strategy for genetic control.

## Results

### Assessing Cas9 and LbCpf1 activity of transgenic strains

Due to the inadequacies of the Gal4/UAS system for directing transgene expression in later stages of spermatogenesis [10], we first generated multiple strains by random or site-directed transgenesis where the *cas9* or *cpf1* (*cas12a*) coding sequences were placed under the direct transcriptional control of the *βtub85D* promoter. This regulatory element drives high expression of genes exclusively in the male germline, during the primary spermatocyte stage (at the onset of Meiosis I) and, as demonstrated in the mosquito [4, 5], this is the stage with the strongest evidence of X-shredding activity. To evaluate CRISPR function, we first crossed *βtub85D*-endonuclease bearing strains to flies containing a gRNA transgene targeting the X-linked, single copy *white* gene, used here as a phenotypic marker (gRNAs *w_ex3_2* for Cas9 and *w_ex3_1* for Cas12a/LbCpf1 were used respectively) [11]. The mutation of *white* results in individuals with white eyes due to a lack of pigment and we compared the activity of the different lines by counting the fraction of offspring with white eyes (S1 Fig, S1 Table). In the progeny of the transheterozygotes we found variability in mutation frequencies ranging from 25.5% with *βtub85D-cas9*[20D] (on the X chromosome) to 52.2% with *βtub85D-cas9*[20F] (on the third chromosome) likely due to differing expression levels of the endonuclease transgene. In addition, the level of activity of all the *βtub85D* driven lines was significantly lower when compared to *cas9* driven by the *nanos* (*nos*) promoter using the same gRNA (96.7% white eyes). LbCpf1 showed low levels of activity (0.6% white eyes) and for our further experiments we exclusively utilized Cas9. Unless specifically indicated, we used the *βtub85D-cas9*[20F] insertion on chromosome 3, which yielded the highest *white* mutation frequency of 52.2% for all experiments.

### Identification of X-linked target sequences

To identify X-linked sequences which could be targeted by CRISPR/Cas9 in the male germline we used two different approaches. First, using publicly available short read and long read *Drosophila* datasets, we employed the Redkmer pipeline [12] which we previously developed to identify putative X-linked repeat sequences that were not also present on other chromosomes from raw sequence data alone (S2 Fig). From the set of kmers of 25 nucleotides that passed our criteria including abundance, X-linkage, and the lack of predicted off-target cleavage, we selected 8 target sequences for gRNA design located in multiple sites across the X-chromosome (S2 and S3 Figs). These target repeat sequences were predicted each to be confined to single chromosomal regions with the 5 most abundant sequences all located within annotated genes (*esi-2.1*, *muc14a*, *hydra*, *CG33235* and *CG15040*). We made no assumption about the higher-order structure or the conservation of these putative repeat sequences between individuals, although all chosen gRNA repeat targets were also predicted to be multicopy sequences on the X when we searched the DmelR6.01 genome assembly (Fig 1B) and all mapped exclusively to scaffold X1 of a recent heterochromatin-enriched genome assembly [13] (S2 Table). The maximum predicted abundance of the top repeat sequences by the assemblies (155 hits) or redkmer (232 hits) suggests that the target repeats in *Drosophila* would be less repetitive than the target in *Anopheles*, i.e. the 28S rDNA cluster with an array in excess of 500 repeats per genome. In a

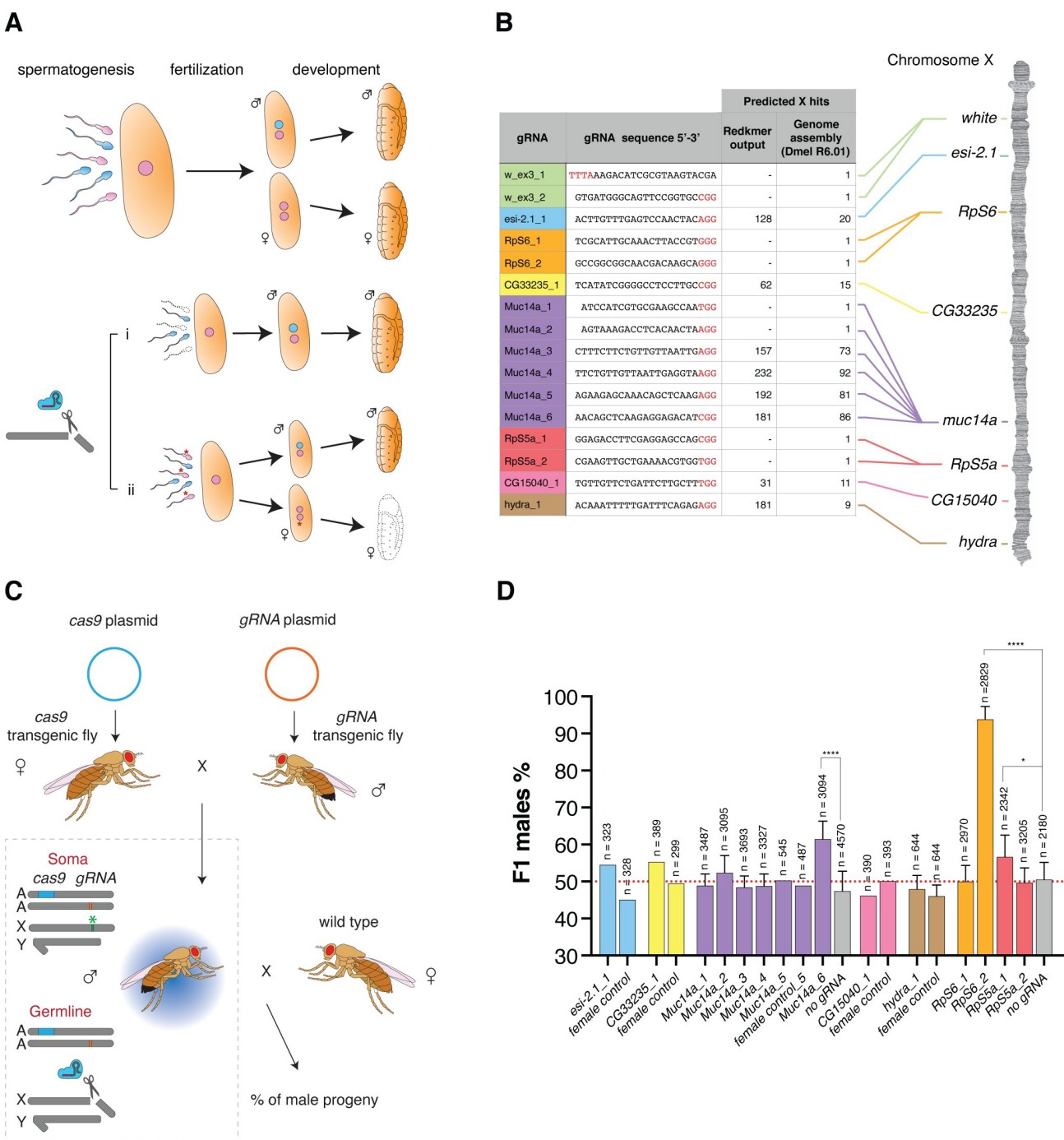

**Fig 1. Development of sex ratio distorters in *Drosophila* (A)** Model for pre-zygotic and post-zygotic effects on the reproductive sex-ratio. Compared to the unaltered process of development (top), an endonuclease targeting the X-chromosome during spermatogenesis (bottom) could either negatively affect the production or function of X-bearing sperm (i, prezygotic effect) or introduce genetic modifications that are detrimental to females inheriting such modified X-chromosomes during development (ii, postzygotic effect; e.g. the mutation of haploinsufficient genes). Pink: X-chromosome bearing gamete. Cyan: Y chromosome bearing gamete. The asterisk indicates gametes bearing modifications at the target site. **(B)** gRNA target sequences used in this study and their positions on the X-chromosome. The protospacer-adjacent motif (PAM) is indicated in red, and the X-chromosome hits predicted by Redkmer and the number of predicted perfect BLASTn hits to the X-chromosome in the *Drosophila* genome assembly (Dmel R6.01) are indicated. **(C)** Schematic of the experimental crosses. Females bearing the *cas9* transgene were crossed to males carrying the *gRNA* transgene. Trans-heterozygote males were then crossed to wild-type females. Cas9 in concert with the gRNA cleaves the X-chromosome at the target site (green asterisk) in the germline and the effect is measured as the sex-ratio of the progeny. A: autosome; X: X-chromosome, Y; Y-chromosome. **(D)** Male sex-ratios in the offspring from crosses of *βtub85D-cas9/gRNA* with wild-type *w* females. Progeny of *βtub85D-cas9/+* males crossed to wild type females (no gRNA) or from the reverse

cross (*βtub85D-cas9/gRNA* females crossed to wild type males = female control) served as a control. Crosses were set as pools of males and females or as multiple male single crosses in which case error bars indicate the mean ± SD for a minimum of ten independent single crosses. For all crosses n indicates the total number of individuals (males + females) in the F1 progeny counted. P-values *p < 0.05, ****p < 0.0001. All crossing data can be found in S3 Table.

second approach, we identified putative haploinsufficient genes [14], on the *Drosophila* X-chromosome and designed four gRNAs targeting conserved regions within the *RpS5a* [15] and *RpS6* [16] ribosomal protein genes.

## Characterization of sex distorting gRNAs

All gRNAs were cloned downstream of the RNA Polymerase III promoter of the *Drosophila* U6 snRNA gene that drives ubiquitous expression and we generated independent strains for all selected gRNAs. In addition, we also generated lines that combined either 2 different gRNAs using double dU6 promoters or 4 gRNAs expressed as a single transcriptional tRNA-gRNA array [17, 18]. The *βtub85D-cas9* and each gRNA line were crossed to obtain trans-heterozygous males expressing both Cas9 and the gRNA in the germline where activity is expected to occur. Next, these individuals were crossed to wild type females to determine the sex-ratio of their offspring (Fig 1C). As a control, we either used *βtub85D-cas9* alone or the reciprocal cross that, due to the lack of *βtub85D* activity in females, was not expected to express Cas9. Fig 1D (S3 Table) summarizes the results of these experiments performed with a pool of individuals or by crossing single males to three females. We noted that while most gRNAs targeting X-linked repeats did not substantially affect the sex-ratio, the *Muc14a_6* gRNA induced a highly significant male-biased sex-ratio of 61.5% (p < 0.0001). This level of distortion was consistently observed in the follow-up experiments. Of those gRNAs targeting putative X-linked haplolethal genes, two gRNAs, *RpS6_2* and *Rps5a_1*, yielded a significant excess of male progeny with frequencies of 93.8% (p < 0.0001) and 56.6% (p = 0.0112), respectively. We first focused our attention on the *Muc14a_6* gRNA that we hypothesized to induce sex-ratio distortion by the X-shredding mechanism. The *Muc14a_6* gRNA maps to the *Mucin 14A* (*Muc14a*, CG32580), an accessory-gland gene of unknown function that spans 52.6 kilobases and contains extended regions of tandemly repeated sequences (see also Fig 3A). The target repeats fall within the coding sequence of the gene with a predicted 271 nucleotides between individual *Muc14a_6* gRNA target sites. We tested four gRNAs, each targeting different but interspersed repeat sequences in the *Muc14a* gene and found that only one yielded significant sex distortion. *Muc14a_6* gRNA produced an excess of male progeny even if its target was not predicted to be the one with the highest number of hits in the cluster (Fig 1B, S2 Table). To exclude the possibility that a loss of *Muc14a* gene function is the cause of the sex-ratio distortion, we also designed two gRNAs, *Muc14a_1* and *Muc14a_2*, targeting putative non-repetitive regions in the *Muc14a* coding sequence. These experiments showed that the function of the *Muc14a* gene was not responsible for the male-biased sex-ratio observed in the progeny (Fig 1D).

We performed *in-vitro* Cas9 cleavage assays and our results suggested that all gRNAs targeting repeat sequences within the Muc14a locus were active *in-vitro*. For this purpose, we amplified and cloned a fragment of 1395 bp of the Muc14a cluster that contained between 2 and 4 target sites for each of the four *Muc14a* gRNAs (Fig 2A) that served as a substrate for *in-vitro* digestions. We therefore concluded that the observed difference in their effects must be due to (i) different intrinsic or *in-vivo* gRNA performance, (ii) the particular nature or context of the target sequence or (iii) the specific repair outcomes triggered. We next tested combinations of gRNAs targeting the same or different X-linked repeat clusters to evaluate whether these could improve sex distortion rates (Fig 2B; S4 Table). While we observed a modest boost by

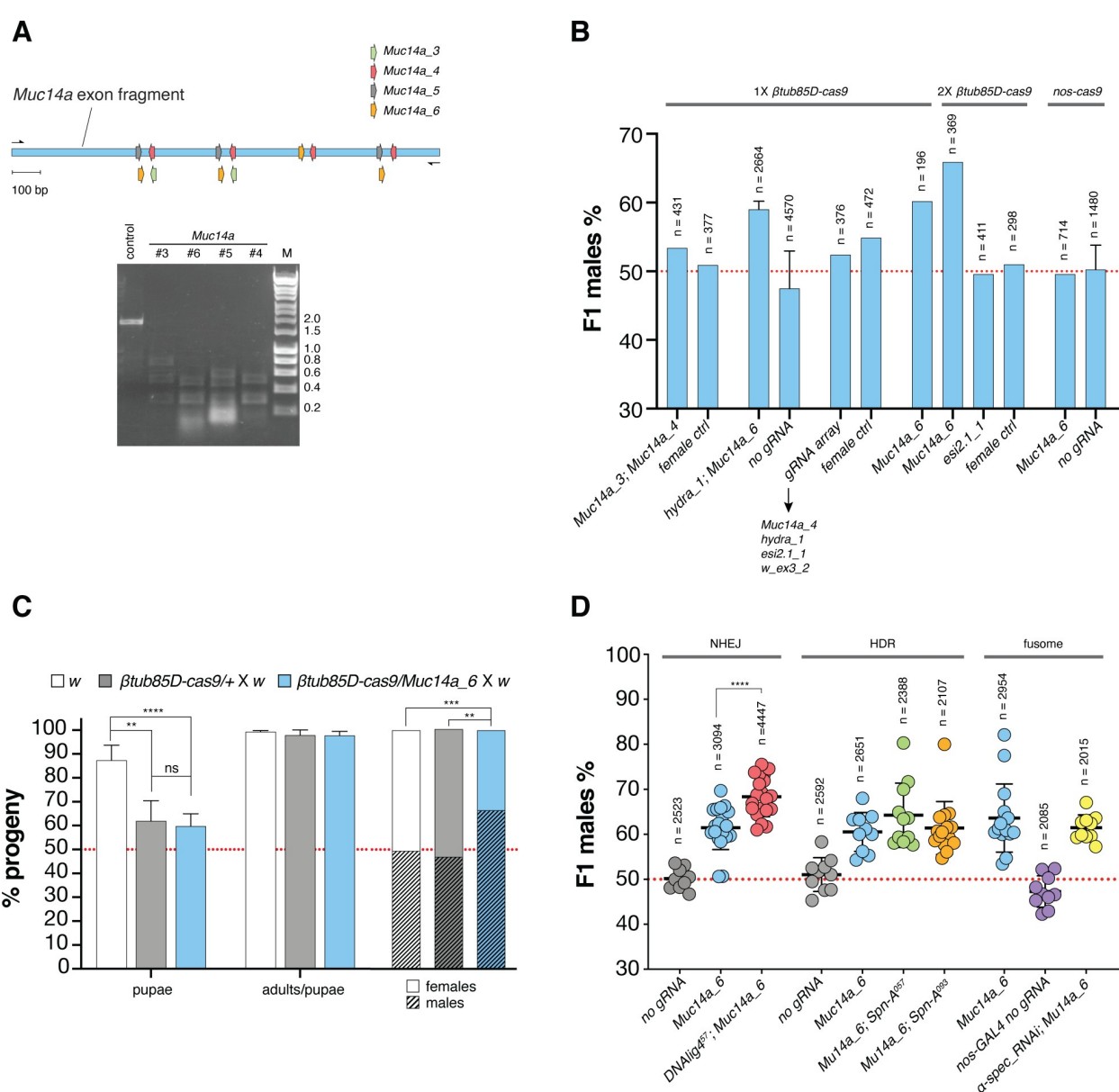

**Fig 2.** **(A)** Assay to detect CRISPR/Cas9-mediated cleavage *in vitro*. The region of the Muc14a gene that was amplified contains at least 2 binding sites for each of the gRNAs: *Muc14a_3*, *Muc14a_4*, *Muc14a_5* and *Muc14a_6* (top). The PCR amplified DNA fragment was used as a digestion target for Cas9/gRNA cleavage reactions *in vitro* (bottom). Reactions were run on a gel to detect cleavage. A control without gRNA was included. **(B)** Analysis of combinations of gRNAs and Cas9 sources for X-shredding. Average male frequencies in the F1 progeny are shown for each parental genotype with a single copy of *βtub85D-cas9* transgene (1X), two copies of *βtub85D-cas9* transgene (2X) or one copy of *nos-cas9* (grey bars). All lines were crossed to wild type *w* individuals. The reciprocal cross (female ctrl) or heterozygote *βtub85D-cas9/+* or *nos-cas9/+* without gRNA (no gRNA) were used as control. The black arrow indicates gRNAs in the multiplex array and the red dotted line indicates an unbiased sex-ratio. Crosses were set as pools of males and females or as multiple male single crosses in which case error bars indicate the mean ± SD for a minimum of ten independent single crosses. For all crosses n indicates the total number of individuals (males + females) in the F1 progeny counted. **(C)** Developmental survival analysis of the F1 progeny of *Muc14a_6/βtub85D-cas9* males crossed to *w* females compared to *w* and *βtub85D-cas9/+* control males crossed to *w* females. Left columns: embryos to pupae survival rate; central columns: pupae to adults survival rate and right columns: fraction of males and females in adults. Bars indicate means ± SD for at least ten independent single crosses. Statistical significance was calculated with a *t* test assuming unequal variance. **p < 0.01, ***p< 0.001 and ****p< 0.0001. **(D)** Influence of DNA repair and fusome integrity on X-shredding. Male frequencies in the progeny of *DNAlig4⁵⁷* (red), *spn-A⁰⁵⁷* (green) and *spn-A⁰⁹³* (orange) mutants and *UAS_α-spectrinRNAi*; *nos-GAL4* (yellow) knockdown mutants. The F1 sex-ratio of the progeny of males carrying *Muc14a_6/βtub85D-cas9* in the mutant backgrounds was compared with those of *βtub85D-cas9/+* (grey), *Muc14a_6/ βtub85D-cas9* (cyan) and *nos-GAL4/+* (purple) control males. Each dot represents the percentage of F1 males from a cross between one male and three females. n is the number of individuals (males + females) in the F1 progeny. Bars show means ± SD for at least ten independent single crosses. Statistical significance was calculated with a *t* test assuming unequal variance. ****p< 0.0001. All crossing data in S3 Table.

providing an additional Cas9 source, neither combination of gRNAs including the transcriptional array of four gRNAs resulted in a higher distortion than that of the single *Muc14a_6* gRNA alone. We also observed that the *w_ex3_2* gRNA targeting *white*, when encoded as the most 3' of four consecutive gRNAs within the tRNA-gRNA array, showed a dramatically reduced level of activity (S1 Fig). In flies expressing both *Muc14a_6* and *hydra_1* gRNAs from separate loci we observed comparable distortion to that of *Muc14a_6* gRNA alone. Interestingly, we observed no sex-ratio distortion when *nos-cas9* males, expressing the *Muc14_6* gRNA, were crossed to wild type females. This result is in stark contrast with *nos-cas9* showing substantially higher levels of activity than *βtub85D-cas9* when combined with the *white* gRNA (S1 Fig). As previously hypothesized [4], these findings support the idea that X-shredding is dependent on the timing of Cas9 activity, with expression in early spermatocytes also working well in the mosquito system. An analysis of *Drosophila* development in the progenies of *βtub85D-cas9/+*, *βtub85D-cas9/Muc14a_6* and wild type fathers confirmed that the *Muc14a_6* gRNA induced no significant zygotic lethality—over any fitness effect of *βtub85D-cas9* alone—an outcome expected if the loss of females is due to X-shredding acting pre-zygotically (Fig 2C; S5 Table).

## X-shredding in backgrounds with impaired DNA repair activity or intracyst communication

To better understand the cellular mechanisms that influence the outcome of X-shredding, i.e. loss of X-bearing gametes and a bias towards males in the progeny, we generated lines bearing mutations in DNA repair pathway components or core components of the fusome. The *DNA Ligase IV* gene (*lig 4*; CG12176) encodes an ATP-dependent DNA ligase involved in non-homologous end joining (NHEJ) DNA repair; *spindle-A* (*spn-A*; CG7948) is a *Drosophila* homolog of the Rad51 gene required for double-strand break (DSB) repair by homologous recombination (HR) in both somatic and germ cells and α-*spectrin* (α-*spec*; CG1977) encodes for a component of the fusome, an organelle that facilitates intracyst cell communications during spermatogenesis [19–21]. The fusome has also been implicated in the coordinated intracyst cell death response following DNA damage and mediating protein transport and diffusion between connected sperm cells, a process on which X-shredding in hemizygote individuals may rely on [4, 21]. We found that the disruption of NHEJ repair by the mutant *lig4*[57] significantly increased the level of male-bias to above 68%, whereas *spn-A* mutants and α-*spec* dsRNAi did not affect the sex-ratio significantly (Fig 2D; S6 Table).

## Analysis of the Muc14a cluster by amplicon sequencing

To gain a more accurate understanding of the DNA repair mechanisms acting during X-shredding we performed sequencing of the *Muc14a* repeat cluster before and after it had undergone modification by Cas9. We crossed the *Muc14a_6* gRNA strain to both, *βtub85D-cas9* and *nos-cas9* flies to examine the outcomes of activity at different stages of spermatogenesis. However, since F1 females inherit modified X-chromosomes from their fathers and unmodified X-chromosomes from their mothers, it is difficult to assess the exact level of gene editing on paternal X-chromosomes. To overcome this obstacle, we crossed *βtub85D-cas9/Muc14a_6* or *nos-cas9/Muc14a_6* male individuals to X^X/Y females with attached-X-chromosomes [22]. By doing this, single modified X-chromosomes are passed from fathers to sons and can be analysed by amplicon sequencing (Fig 3A). In control males, ~90% of repeats contained the full gRNA target site (Fig 3B, panel 1) although polymorphisms in the surrounding sequence indicated that the Muc14a cluster shows substantial heterogeneity. We found that, on average, X-chromosomes inherited from *βtub85D-cas9/Muc14a_6* fathers showed a > 40% reduction of cleavable gRNA target sites (Fig 3B, panel 1) but surprisingly this value was over 85%, on average, in

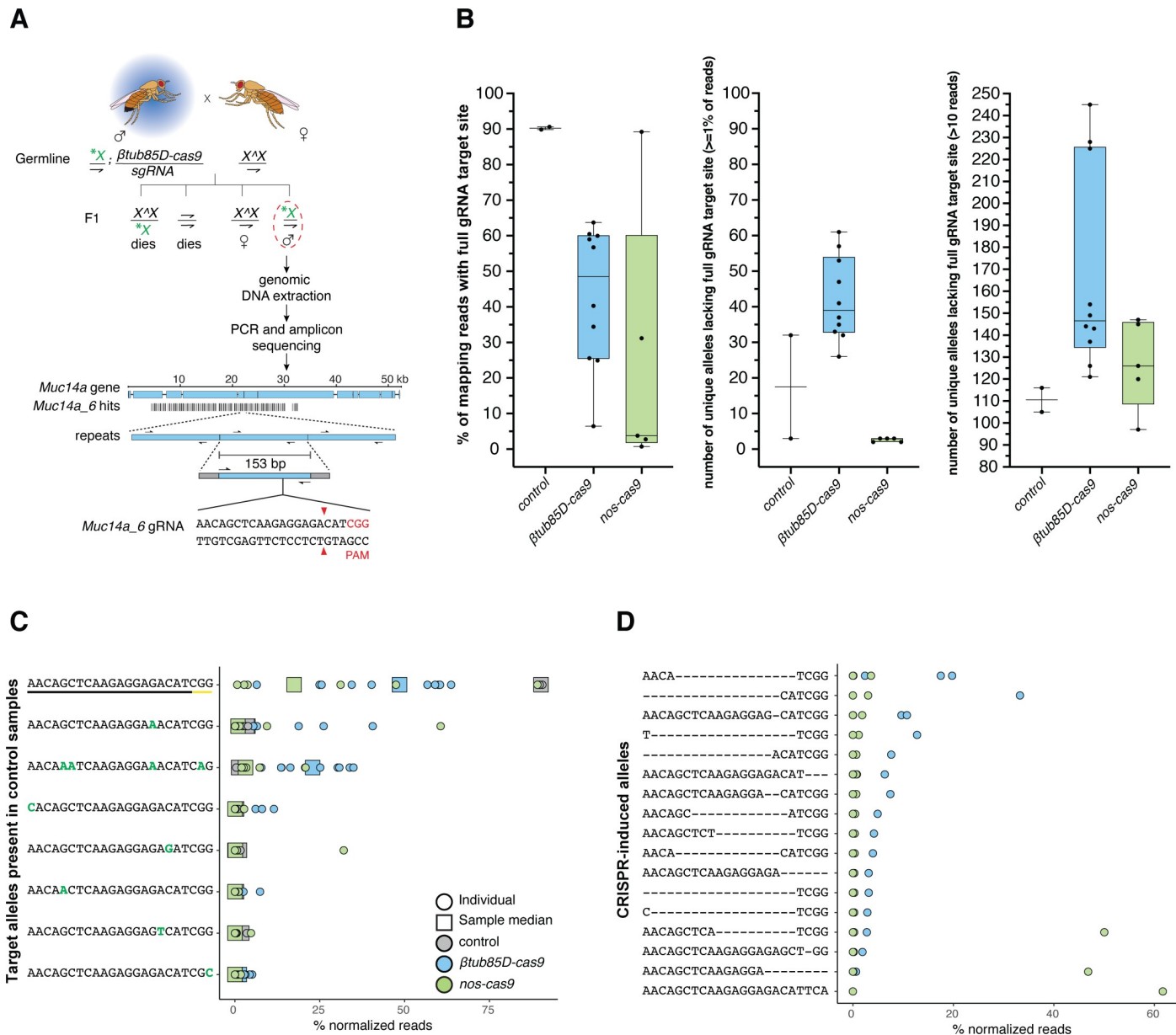

**Fig 3.** (**A**) Schematic of the genetic crosses to obtain shredded X-chromosomes from *Muc14a_6/βtub85D-cas9* or *Muc14a_6/nos-cas9* fathers for amplicon sequencing. Trans-heterozygous males for *cas9* and the *gRNA* were crossed to females with attached-X-chromosomes (X^X). Offspring with supernumerary or lacking X-chromosomes are inviable leaving X^X/Y females and males carrying the X-shredded chromosome (patroclinous inheritance) which we selected and analysed. Genomic DNA from single males was extracted and a 153 bp DNA motif containing the *Muc14a_6* gRNA target site was amplified with primers (thin arrows on repeats) containing Illumina Sequence adapters (grey boxes flanking the amplicon). As a comparison we used DNA from *βtub85D-cas9*/+ individuals lacking the gRNA. The shredded X-chromosome is indicated by a green asterisk. A dashed red circle indicates males selected for amplicon sequencing. Vertical black bars represent the number and location of the *Muc14a_6* gRNA repeats within the *Muc14a* gene. (**B**) Analysis of allele variation at the *Muc14a_6* target site by amplicon sequencing. Indicated is the percentage of all mapped reads that contain the complete, unaltered *Muc14a_6* target site (left panel) in the control and experimental males. The middle and right panel show the number of reads harbouring all other unique alleles that represented at least 1% of all reads or were represented by 10 or more reads, respectively. (**C**) Analysis of unique alleles (including the wild type target site) at the *Muc14a_6* target site that pre-existed in both control samples. Indicated are the relative frequencies of these alleles in each control and experimental male (circles) including the median frequency of each allele in all *Muc14a_6/βtub85D-cas9* or *Muc14a_6/nos-cas9* males (squares). (**D**) Analysis of *de-novo* alleles at the *Muc14a_6* target site not present in control males and putative CRISPR-induced alleles. In C and D we considered only alleles that represented ≥1% of normalized reads in at least one male sample.

those X-chromosomes derived from *nos-cas9/Muc14a_6* males, despite the fact that no sex-ratio distortion had been observed with this combination. X-chromosomes of *βtub85D-cas9/Muc14a_6* fathers had, on average a larger number of unique alleles (identified from mapping reads that lacked the gRNA at the target site) than the wild type. By contrast, the progeny of *nos-cas9/Muc14a_6* showed a reduction of allele diversity, when considering alleles represented in $\geq$ 1% of mapping reads (Fig 3B, panel 2). A similar pattern was observed when we considered alleles with a read coverage above 10X, where X-chromosomes derived from *βtub85D-cas9/Muc14a_6* males showed a more diverse repeat landscape compared to control and *nos-cas9/Muc14a_6* males (Fig 3B, panel 3). We analysed in more detail the fate of pre-existing alleles in the control males (Fig 3C) as well as novel alleles presumably generated by CRISPR/Cas9 activity (Fig 3D). While consensus allele frequency decreased in males of the experimental groups, we found a pre-existing allele predicted to be cleavage-resistant (AACAaATCAAGAGGAAACATCaG, with mutations in both the target site and the PAM), to increase in median frequency from <1% to ~25% in males from *βtub85D-cas9/Muc14a_6* but not from *nos-cas9/Muc14a_6* fathers (Fig 3C, row 3). In contrast, novel CRISPR-induced alleles, likely generated pseudo-randomly, were commonly shared amongst a few males, although their mapping reads accounted for more than 40% of all reads in some males from *nos-cas9/Muc14a_6* fathers (Fig 3D). On the sequence level, CRISPR-induced alleles consisted mainly of smaller deletions at or around the site of cleavage and all would be predicted to prevent further Cas9 cleavage. Together these findings suggest quite different dynamics of repair or repair mechanisms between the early acting *nos-cas9* and the meiotic *βtub85D-cas9*.

## Targeting putative haplolethal X-linked genes

We next focussed our analysis on the set of gRNAs targeting putative haploinsufficient single-copy genes. As we observed for gRNAs targeting the Muc14a repeats, no combination of gRNAs with *βtub85D-cas9* was found to increase the sex-ratio relative to the *RpS6_2* gRNA alone. Both gRNAs, *RpS5a_1* and *RpS6_2*, when crossed to *βtub85D-cas9* were individually capable of inducing male-bias sex-ratio distortion, but when combined within a gRNA array, we found a significantly lower level of distortion in the progeny than when we used *RpS6_2* alone (Fig 4B; S7 Table). In contrast to X-shredding, X-poisoning is expected to act post-zygotically by inducing female lethality in the developing embryo likely as a result of an insufficient dose of the target-gene product. We examined *Drosophila* development in the progenies of *βtub85D-cas9/+*, *βtub85D-cas9/RpS6_2* and wild type fathers and, in contrast to the *Muc14a_6* gRNA, we did observe significant lethality at the embryonic and subsequent stages. This result would suggest a mechanism of lethality that operates during development and that the few survivor females in the progeny carried rescue mutations that restored the function of the ribosomal target genes, while preventing further CRISPR cleavage. To confirm this hypothesis, we performed amplicon sequencing of the *RpS6_2* target sequence using 2 pools of the surviving females from *βtub85D-cas9* fathers expressing the *RpS6_2* or *RpS6_2* and *RpS6_1* gRNAs (Fig 4C; S8 Table). Surprisingly, while we did observe possible rescue alleles, the majority of mutant alleles identified in surviving females are not expected to restore *RpS6* gene function because of the presence of translational frameshifts (Fig 4D). Although, one has to be cautious to infer loss of function from such experiments alone [23], these results imply that the mechanism of lethality following *RpS6* cleavage and imperfect repair is dominant rather than dependent on an insufficient gene dose. Co-expression of *Muc14_6* and *Rps6_2* gRNAs, i.e. combining the two mechanism of distortion we have described, yielded the highest level of distortion we observed in this study (on average 95.8% males) although this was not significantly different from *RpS6_2* alone. Finally, when we combined the *Rps6_2* gRNAs with *nos-cas9* the result

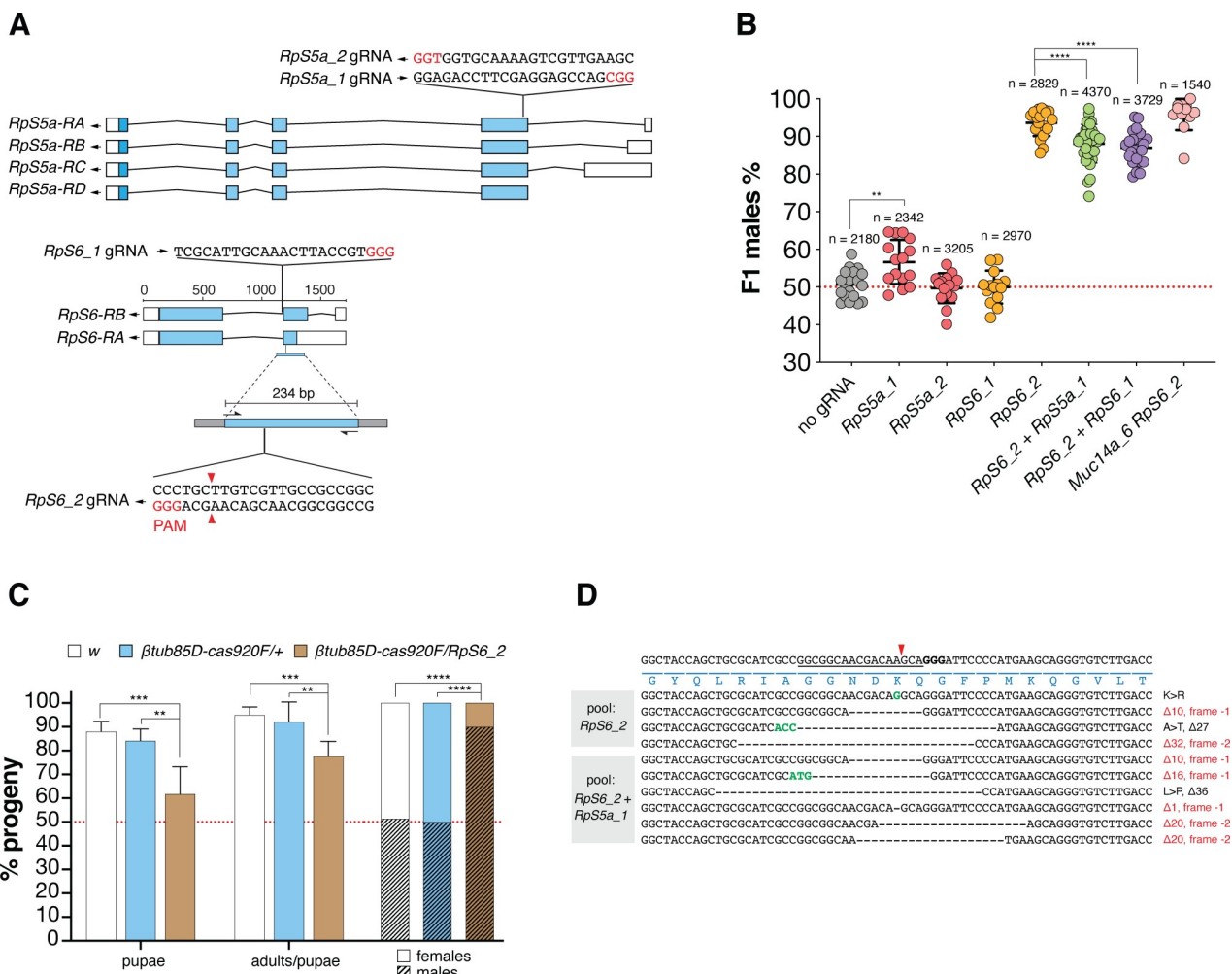

**Fig 4.** **(A)** gRNA target sites within presumptive haploinsufficient genes on the X-chromosome. Schematic representation of *RpS5a* and *RpS6* gene organization. Both genes encode for a small ribosomal subunit protein (RpS). As illustrated in the figure, the *RpS5a* gRNAs, *RpS5a_1* and *_2*, map in the fourth exon shared by all four transcripts of the gene and the *RpS6* gRNAs, *RpS6_1* and *2*, map in the third exon of two transcripts in the corresponding gene. The figure shows the 234 bp fragment (with Illumina Sequence adapters on both sides as grey boxes) surrounding the *RpS6_2* gRNA target site that was used for amplicon sequencing. Blue boxes indicate coding sequences, white boxes indicate UTR regions and PAMs are indicated in red. **(B)** Efficiency of single gRNAs or combinations of gRNAs for X-poisoning. Shown is the frequency of males in the progeny from *βtub85D-cas9* males combined with four *gRNA* lines and crossed to wild type *w* females. Individual gRNAs, *RpS5a_1* or *RpS5a _2* (red), and *RpS6_1* or *RpS6_2* (orange) are compared to double gRNA arrays co-expressing *RpS6_2* + *RpS5a_1* (green) or *RpS6_2* + *RpS6_1* (purple) as well as a combination of both *RpS6_2* and *Muc14a_6* transgenes (pink). As a control, crosses from *βtub85D-cas9/+* (no gRNA, grey) fathers are shown. Each dot represents the percentage of F1 males from a cross between one male and three females. n is the number of individuals (males + females) in the F1 progeny. Black bars show means ± SD for at least ten independent single crosses. Statistical significance was calculated with a *t* test assuming unequal variance. **p < 0.01, ***p< 0.001 and ****p< 0.0001. All crossing data can be found in S7 Table. **(C)** Developmental survival analysis of the F1 progeny of *RpS6_2/βtub85D-cas9*, *w* or *βtub85D-cas9/+* males crossed to *w* females. Left columns: embryos to pupae survival rate; central columns: pupae to adults survival rate and right columns: fraction of males and females in adults. Bars indicate means ± SD for at least ten independent single crosses. Statistical significance was calculated with a *t* test assuming unequal variance. **p < 0.01, ***p< 0.001 and ****p< 0.0001. All crossing data can be found in S8 Table. **(D)** CRISPR induced target site mutations within the *RpS6* gene analysed by pooled amplicon sequencing of surviving female progeny. On top, the wild type DNA sequence spanning the *RpS6_2* gRNA target site in the *RpS6* ribosomal gene is shown with the gRNA binding position (underline), the cut site (red arrowhead), the PAM (bold nucleotides) as well as the encoded amino acids (blue). Target site variants identified in pools of F1 females from *RpS6_2/βtub85D-cas9* fathers or *RpS6_2 RpS5a_1/βtub85D-cas9* fathers are shown below. Dashed lines correspond to nucleotide deletions, green coloured bases represent insertions. Predicted amino-acid substitutions (>) are indicated to the right.

was complete sterility of the transheterozygous males (Supplementary Dataset 1). Since ribosomal gene function is required during gametogenesis, disrupting *RpS6* during the stem cell stages appears to be detrimental for sperm development.

## Discussion

To recreate in *Drosophila* a synthetic X-shredding mechanism pioneered in *A. gambiae*, we generated gRNAs targeting repeat sequences we identified on the X-chromosome. This demonstrates that X-shredding is a transferable mechanism and not linked to the particular nature of the mosquito target, i.e. the ribosomal DNA gene cluster that was targeted in previous studies [5, 6]. It also shows that CRISPR/Cas9 target sequences that are able to induce sex-ratio distortion can be identified bioinformatically. To search for such X-linked repeats, we employed the Redkmer pipeline using raw sequence data as the only input. This is crucial as many target species of medical or agricultural importance are likely to lack high-quality genome assemblies. Even when such assemblies exist, repeats sequences, a problematic class of sequences for assemblers, often remain poorly resolved. However, recent progress in telomere-to-telomere chromosome assemblies that incorporate large DNA repeat clusters may simplify this step in the future [24]. The level of sex distortion towards males we observed in the fly was not as extreme as observed in *A. gambiae*. Given that activity against the single-copy *white* was incomplete this suggests possible improvements by further optimizing the level of *cas9* expression in the germline of *Drosophila*. Using our best *cas9* strain, we found that achieving higher rates of distortion required us to interfere with mechanisms of DNA repair. This in turn indicates that targeting more repetitive sequences could be another avenue to enable a more dramatic bias towards males, though the *Drosophila* X-chromosome lacks such repeats. The smaller repeat unit size and hence smaller distance between individual cut sites (271 base pairs in the case of *Muc14a_6* versus ~9kb in the *A. gambiae* rDNA target) could also have increased the chances of successful repair. In addition to the intrinsic activity of each gRNA, the sequence microenvironment rather than the number of repeats may also play a role in determining why certain gRNAs trigger gamete loss while others don't. The identification of X-chromosomes which are more suitable for X-shredding in target species of medical or agricultural relevance is thus the next task. Our study provides lessons for the application of X-shredding to such species. A strong meiotic promoter yielding high levels of Cas9 should be combined with as few gRNAs as possible, ideally one, targeting a highly repetitive sequence, which does not have to be essential and can consist of a single repeat cluster on the X-chromosome.

Chromosome-wide, distributed repeats represent an alternative set of targets but the lack of such sequences in *Drosophila* precluded us from evaluating their use for X-shredding. The fly literature suggests that the 1.688 X-chromosome satellite involved in X dosage compensation [25] one of the most abundant repetitive sequences in *Drosophila melanogaster*, would represent such a target [26]. While the set of candidate targets did include a number of kmers that could be attributed to the 1.688 satellite they did not pass our selection criteria or were significantly less abundant than the targets we selected. This is likely due to the heterogeneity and stratification of 1.688 repeats in various chromosomal locations [27].

Although the use of larger arrays of gRNAs has been suggested to compensate for the lack of abundant X-linked repeats [6], our data suggests that targeting a highly repetitive sequence with a single gRNA may be a more viable route. Even if multiple gRNAs could be expressed efficiently and concomitantly, they could compete for access to Cas9 protein in the loading step thus reducing activity of each individual gRNA. The saturation of Cas9 by gRNAs has been observed previously [28].

Our data suggests that the high level of activity of *nos-cas9* with *Muc14a_6* gRNA during early spermatogenesis could cause the repeats to be subject to multiple cleavage-repair cycles which in turn may lead to the observed reduction in the complexity and presumably also the size of the repeat cluster (and no sex bias). By contrast, *βtub85D-cas9/Muc14a_6* activity, which acts later in meiosis, would encounter, following multiple mitotic divisions, more X-

chromosomes to target in the larger population of primary spermatocytes on which it may be acting continuously and in parallel to generate novel alleles. This would result in a broader spectrum of mutations inherited by the progeny. Alternatively, the observed differences could also partly relate to the predominance of homologous over non-homologous DNA repair pathways acting with varying stringency during the early (including stem cells) and late stages of spermatogenesis, respectively [29] or an overall greater repair efficiency in pre-meiotic stages compared to later stages when *βtub85D-cas9* is active. While *nos-cas9* would favour the loss of repeat units by recombination-based repair mechanism, *βtub85D-cas9* would trigger NHEJ repair events leading to a more diverse allele pool. Indeed, we found little evidence of homologous repeat to repeat repair in *βtub85D-cas9/Muc14a_6* males. For instance, the pre-existing cleavage-resistant allele AACAaATCAAGAGGAAACATCaG is associated with a 6bp indel polymorphism located 53 nucleotides upstream of the gRNA target. We did not detect dissociation of this SNP even in samples in which the cleavage resistant allele rose to a frequency of 30% of the mapped read pool. Dissociation would have indicated that a double-strand break in a wild-type repeat unit had been repaired by gene conversion using the cleavage resistant allele as a template.

One caveat is the fact that the X-chromosomes from *βtub85D-cas9/Muc14a_6* males we analysed by amplicon sequencing managed to escape the germline in the form of X-bearing gametes and gave rise to viable males. One might argue that they represent chromosomes with an overall lower level of modifications or modifications with repair outcomes compatible with transmission and male survival (which may in turn differ from the requirements for female survival). In contrast, the X-chromosomes unable to form viable gametes, i.e. those that we could not analyse by sequencing the surviving progeny, may be the ones that underpin the sex distortion observed, and may be subject to fundamentally different repair events.

A number of questions remain to be answered, such as the mechanism through which X-shredding causes the loss of X-bearing gametes and how insufficient or incomplete DNA repair is associated with this process. This outcome is by no means self-evident; for example, it has been shown that functional sperm can be produced despite lacking either one or both major autosomes or lacking DNA [30]. The fly model of X-shredding we have established will allow to tackle these questions experimentally. Much remains to be understood also before X-shredders could move towards application. For example, CRISPR-induced X chromosome rearrangements or pathways for the evolution of cleavage-resistant repeats are areas that have not been sufficiently explored.

We also established an X-poisoning system in the *Drosophila* model by targeting X-linked ribosomal genes and achieved high rates of male bias. This latter strategy, in the form of a Y-linked endonuclease, has recently been proposed as an efficient alternative to gene drives for genetic control [8]. Modelling suggests Y-linked editors would outperform other self-limiting strategies while having less impact on non-target populations when compared to gene drives or driving Y-chromosomes. We found the mechanism of embryo lethality in the case of the ribosomal *RpS6* gene to be more complex than anticipated. Rather than protein dose insufficiency, a dominant lethal effect may partially or totally explain our results. Such an effect in mutated ribosomal proteins has been observed previously [31] and could, for example, be explained by the poisoning of ribosomes with dysfunctional ribosomal proteins. Since, in this case, every escaper female represents only a single repair event, phenotypes would be expected to vary depending on the outcome of DNA repair. Another possibility to explain the survival of females with frameshift mutations in the target gene is that the haplolethality of RpS6 could be time-dependent so that after passing a "critical" stage of development, e.g. embryo development, the gene becomes haplosufficient. Further modelling may be required to understand how these phenotypes would impact genetic control at the population level.

The X-shredding and the X-poisoning transgenes we have described here could be expressed from the *Drosophila* Y chromosome which has recently been modified to harbour pre-characterized gRNA sites for transgene insertion [32]. This could allow, for the first time, to explore the use of such Y-linked editors with gRNAs for X-poisoning or X-shredding. To what degree these two strategies could be designed around or would be susceptible to X-chromosome inactivation in the male germline of *Drosophila* is a further research question, in particular as this process is not fully understood in the fly [33, 34]. In both strategies timing of expression appears key and too early expression—when Cas9 was driven by the *nos* promoter —was unsuccessful, resulting in no distortion for the X-shredding and male sterility for the X-poisoning strategy.

Nevertheless, the application of X-poisoning to agriculturally or medically relevant species would appear to be a more straightforward proposition in particular as ribosomal target genes show high levels of conservation and because lower levels of Cas9 activity for cleaving a single target site per chromosome may suffice. Also, the characterization of Y chromosomes and the identification of male determining regions [7, 35, 36] and docking sites [32] within which to land transgenes has recently made great progress.

## Materials and methods

### Identification of candidate kmers for X-shredding

Error-corrected PacBio reads derived from males of the ISO1 strain (ref doi: 10.1038/nbt. 3238) were obtained from the University of Maryland Center for Bioinformatics and Computational Biology (http://gembox.cbcb.umd.edu/mhap/data/dmel.polished.fastq.gz). Illumina reads from males and females of the same strain were obtained from the NCBI short read archive (female: SRX826515 and male: SRX826516). These were used as inputs to run Redkmer [12]. The mitochondrial genome (NCBI accession number: NC_024511) was used to filter out mitochondrial derived reads. We only considered Pacbio reads between 2kb and 100kb to improve chromosome quotient accuracy (CQ–[37]) and excluded kmers occurring less than 4 times in the combined male and female Illumina data. We applied a number of critera to the 25,298 candidate X-kmers predicted by Redkmer to be X-linked abundant sequences. First, we used FlashFry [38] to select among the candidate X-kmers, those that contained a sequence suitable for the design of a Cas9 gRNA. To estimate off-target potential, we used as a reference all PacBio reads predicted by Redkmer to represent autosomal and Y derived sequences. We then applied several additional metrics such as the maximum number of occurrences of a kmer per PacBio read, giving an estimate of the repeat size, and the total number PacBio reads containing the kmer at least once, giving an estimate of the abundance of the repeat across the X-chromosome. Kmers shortlisted for testing were selected based on the following cutoffs: abundance in the genome (coverage in combined male and female Illumina data $>\log_{10}(2)$, coverage in PacBio data $>\log_{10}(1)$, the maximum kmer hits per PacBio read $> 7.5$, and at least 5 perfect blast hits to the assembled genome. All candidate X-kmers were also re-assembled into longer contigs using the Geneious Software (Geneious Assembler allowing no mismatches) to identify possible higher-order repeat loci and to ensure that different sequence classes would be targeted in our experiments. A total of 8 of 205 X-kmers passing all filters were then selected for testing in transgenic strains.

### Generation of transgenic lines

**Design and assembly of constructs.**   Unless otherwise noted, cloning was performed with the NEBuilder Hi Fi DNA Assembly kit (New England Biolabs). PCR reactions were performed with the Phusion High-Fidelity PCR Master Mix with HF buffer (New England

Biolabs). All inserts were verified by sequencing (GENEWIZ). Primers used for plasmid construction are listed in S9 Table.

**βtub85D-cas9 expression plasmid.**   The previously described plasmid *pYSC47w⁻* harbouring the *3Px3-eGFP* transformation marker, *attB* and *piggyBac* recombination sequences (a gift from Andrea Crisanti, Imperial College London) was used to build *Drosophila cas9* constructs. The plasmid was linearized with *Sbf*I. A 478 bp DNA fragment of *βtub85D* 5' regulatory region was amplified with primers *βtub85D F/βtub85D-cas9-R* from the genomic DNA of *Drosophila* strain $w^{1118}$. The human codon-optimized *cas9* coding sequence including two nuclear localization signals (*SV40 NLS* at the 5' and *nucleoplasmin NLS* at the 3') was amplified from *hcas9* (a gift from George Church; Addgene plasmid # 41815; http://n2t.net/addgene:41815; RRID: Addgene_41815) using primers *βtub85D-cas9-F/cas9-βtub56D-3'UTR-R* [39]. A third DNA fragment of 651 bp in the *βtub56D* 3'-UTR region was amplified with primers casβtub56F/ βtub56D R from the plasmid *pYSC61*. The three fragments were cloned in the open backbone of *Sbf*I-linearized *pYSC47w⁻* resulting in the final plasmid termed "*pYSC47w+_βtub85D_-cas9_βtub56D*". The plasmid was used for both *piggyBac* and *attP*-docking site integration.

**βtub85D-Lbcpf1 expression plasmid.**   To generate the plasmid expressing *Lbcpf1*, fragments containing the *βtub85D* promoter and the *βtub56D* 3'-UTR were amplified from the construct *pYSC47w+_βtub85D_cas9_βtub56D*. The fragment containing the *Lbcpf1* coding sequence was amplified from *pY016 hLbcpf1* (Addgene plasmid # 69988; http://n2t.net/ addgene:69988; RRID:Addgene_69988) [40]. The three fragments were finally assembled in *Sbf*I-digested *pYSC47w⁻*.

**Guide RNA expression plasmid.**   All single guide RNAs were cloned in *Bbs*I-linearized *pCFD3-dU6:3 gRNA* plasmid (a gift from Simon Bullock; Addgene plasmid # 49410; http://n2t. net/addgene:49410; RRID:Addgene_49410) [17] and harbouring a U6 promoter. A gRNA array containing *Muc14a_4*, *hydra_1*, *esi2.1_1* and *w_ex3_2*, in this order, was assembled with the oligos *Array1_cas9_PCR1_F*, *Array1_cas9_PCR1_R*, *Array1_cas9_PCR2_F*, *Array1_ cas9_PCR2_R* and *Array1_cas9_PCR3_R* and cloned in *pCFD5* plasmid (a gift from Simon Bullock; Addgene plasmid # 73914; http://n2t.net/addgene:73914; RRID:Addgene_73914) [18]. Two distinct arrays, each containing two gRNAs targeting ribosomal genes, were assembled with primers *RpS6 2_pCFD4_F* and *RpS6_ 1_pCFD4_R* for the *RpS6_2 + RpS6_1* array and *RpS6_2_pCFD4_F* and *RpS5a_1_pCFD4_R* for the *RpS6_2 + RpS5a_1* array. The arrays were cloned in *pCFD4-U6:1_U6:3* tandem gRNAs plasmid (a gift from Simon Bullock; Addgene plasmid # 49411; http://n2t.net/addgene:49411; RRID:Addgene_49411). The plasmids *pCFD3*, *pCFD4* and *pCFD5* were all integrated in *attP*-docking sites. CRISPR target site design. CHOPCHOP v2 was used to choose gRNA target sites in *RpS6* and *RpS5a* ribosomal genes specific regions in *Drosophila* genome (dm6).

**Embryo injections.**   Embryo injections were carried out at the University of Cambridge Fly Injection Facility. The *βtub85D_cas9* constructs were inserted at the *P{CaryP}attP40* site on the 2nd chromosome (25C6; Stock 13–20) and the *PBac{y⁺-attP-9A}VK00027* on the 3rd chromosome (89E11; Stock 13–23), both stocks marked with *yellow⁺*. Moreover, the *βtub85D_cas9* plasmid was integrated in *D. melanogaster* by piggyBac transposition. Transgenic flies were balanced with *w¹¹¹⁸; if/CyO* and *w¹¹¹⁸; TM3, Sb/TM6B*. Different lines were generated with *βtub85D_cas9* integrated on the X (20D), on the second (20G) and on the third chromosome (20F). gRNAs cloned in *pCFD3*, *pCFD4* or *pCFD5* were integrated in the genome at the *P{CaryP}attP40* site on the second (Bloomington stock 25709) and/or *P{CaryP}attP2* site on the third chromosome (Bloomington stock 25710). *pCDF4_RpS6_2_RpS6_1*, *pCDF4_RpS6_2_RpS5a_1*, *pCFD3_w_ex3-2*, *pCFD3_RpS6_gRNA2*, *pCFD3_RpS6_gRNA1*, *pCFD3_Muc14a_1*, *pCFD5_Cas9_Array1*, *pCFD3_CG33235_1*, *pCFD3_CG15040_1*, *pCFD3_Hydra_1*, *pCFD3_Muc14a _4*, *pCFD3_ Muc14a_3*, *pCFD3_ Muc14a_3*, *pCFD3_*

*Muc14a_5*, *pCFD3_ Muc14a_6*, and *w_ex3-1* (LbCpf1) were inserted in the *P{CaryP}attP2* site on the third chromosome and *pCFD3_RpS5a_gRNA1*, *pCFD3_RpS5a_gRNA2*, *pCFD3_ Muc14a_2*, *pCFD3_Hydra_1*, *pCFD3_ Muc14a_3*, *pCFD3_ Muc14a_5*, and *pCFD3_ Muc14a_6* were inserted in the *P{CaryP}attP40* site on the second chromosome.

## Fly husbandry and strains

Flies were maintained under standard conditions at 25˚C with a 12/12 hour day and night cycle. For general maintenance, stocks were provided with new food every 2–3 weeks. Flies were anesthetized with $CO_2$ during phenotyping. To assess fluorescent green eyes phenotype conferred by the *3PX3::eGFP* transgene, we used a conventional fluorescence microscope. Description of stocks not provided here can be found in FlyBase (http://flybase.org). The lines used in this study were: *DNAlig4[57]* Bloomington Stock #8520, *Spn-A[057]* and *Spn-A[093]* a gift from Mitch McVey (Tuft University), Df(3R)XF3 Bloomington Stock #2352, *UAS-α-spectrinR-NAi* Bloomington Stock #56932 (TRIP.HMC04371), *nanos_Gal4* Bloomington Stock #4937, *C(1)DX/FM6* (compound chromosome with attached X-chromosomes) Bloomington stock #784.

## Analysis of *βtub85D-cas9* and *nos-cas9* activity

To test *attP* or *piggyBac* integrated *βtub85D-cas9* and *nos-cas9* efficiencies, females from each line were crossed to *w⁺; w_ex3-2* gRNA males bearing a gRNA targeting the *white* gene on the X-chromosome. Red eye trans-heterozygous *w⁺; cas9/w_ex3-2* F1 males were selected, crossed to *w^1118* females and the percentage of white eye females in the progeny was recorded.

## Genetic crosses for X-shredding and X-poisoning

During the initial screen for X-shredding-induced sex ratio distortion, *βtub85D-cas9* mothers were crossed to each gRNA line (from a 25 nucleotides kmer), in a vial at 25˚C. At least twenty trans-heterozygous F1 *gRNA/+; βtub85D-cas9/+* males for each combination of sgRNA were then crossed to the same number of *w⁻* females in new vials at the same temperature. We used the reverse crosses, *βtub85D-cas9/sgRNA* females crossed to *w⁻* males, as controls due to the lack of *βtub85D* activity in females. We discarded the gRNA lines that did not result in sex ratio distortion while those where we recorded a male bias progeny were further tested by pursuing single cross analysis. For every single cross, a single trans-heterozygous *βtub85D-cas9/+; gRNA/+* male (experiment) and a heterozygous *βtub85D-cas9/+* male (control) were crossed to three wild-type females in separate vials at 25˚C. We set up a minimum of ten single crosses for each genotype analysed. For all crosses where we analysed the change in male bias in the progeny of *βtub85D-cas9/+; Muc14a_6/+* in a NHEJ, HDR or fusome mutant backgrounds, we set up ten-twenty replicas of single male crosses, i.e., *w DNAlig4[57]; Muc14a_6/βtub85D-cas9[20F]* single male X three *w* females, *Muc14a_6/βtub85D-cas9[20G]; Spn-A[093 or 057]/Df(3R)XF3* single male X three *w* females and *UAS-α-spectrinRNAi/Muc14a_6; βtub85D-cas9[20F]/nanos_-Gal4* single male X three *w* females and compared the results to the progenies of *Muc14a_6/βtub85D-cas9* and *βtub85D-cas9/+* single male crosses used as controls.

In all crosses for X-poisoning-induced sex ratio distortion, *βtub85D-cas9* mothers were crossed to each gRNA analysed, in a vial. Single F1 *βtub85D-cas9/+; gRNA/+* (experiment) and *βtub85D-cas9/+*(control) males were then crossed to three wild-type females in separate vials. We set up a minimum of ten single crosses for each genotype to generate means and standard deviations for statistical comparisons and thus measure consistency and robustness of the results. All crosses were done at 25˚C. Percent of males and females was calculated as the ratio between the number of individuals counted and the number of individuals expected for each

genotype. Flies were scored and examined with the Nikon SMZ1500 stereomicroscope equipped with a CoolLED $p$E-300 Led fluorescent illumination.

### *In vitro* cleavage assay

The *Muc14a_3*, *Muc14a_4*, *Muc14a_5* and *Muc14a_6* gRNAs were tested for *in vitro* activity using the Guide-it sgRNA *In Vitro* Transcription and Screening System (Takara Bio USA, Inc.). Each gRNA was transcribed and purified according to the *In Vitro* Transcription of sgRNA protocol. Two experiments were performed, allowing transcription to take place for 4 and 8 hours. Genomic DNA templates were obtained from Muc14a fragments previously cloned in a *pMiniT 2.0* vector using PCR Cloning Kit (NEB) and primers kmer Muc14_F and 58537687_F. The genomic target was amplified by PCR with Cloning Analysis Forward and Reverse Primers, mapping in the *pMiniT 2.0* plasmid. The amplicon corresponds to a genomic fragment of 1395 bp and sequencing confirmed the presence of target sites for all 4 gRNAs. For *in-vitro* digestions the DNA was excised from an agarose gel and purified with Monarch DNA Gel Extraction Kit (NEB). DNA concentration and absorbance ratio were measured with a NanoDrop Spectrophotometer (ThermoScientific). PCR amplification of target DNA and a Cas9 cleavage assay were then carried out according to the protocol.

### Amplicon sequencing analysis preparation

Genomic DNA was isolated from ten single F1 males originated from the cross *βtub85D_-cas9$^{20F}$/Muc14a_6* X *C(1)DX/FM6* (compound chromosome with attached X-chromosomes) Bloomington stock #784 and from one control male from the cross *w/Y; βtub85D_cas9$^{20F}$/TM6B* using the QIAamp DNA Micro Kit. Genomic loci containing the *Muc14a_6* gRNA target site were amplified with Phusion HF DNA polymerase (Thermo Scientific) using *Repeat 56910823_illumina_F* and *Repeat_Muc14a_3_illumina_R* primers containing the Illumina adapters. 200 ng of genomic DNA in a 100 μl reaction volume were used as a template for a limiting PCR reaction to amplify 153 bp of slightly different repeats in the *Muc14a* gene. To maintain the proportion of reads corresponding to particular repeats, the PCR reactions were performed under non-saturating conditions for a total of 25 cycles with 55˚C annealing temperature.

For deep sequencing analysis of the ribosomal gene *RpS6* target site, the genomic DNA was extracted from a pool of four daughters from the cross *βtub85D_cas9$^{20F}$/RpS6_2* X *w*, four daughters from the cross *βtub85D_cas9$^{20F}$/RpS6_2 + RpS5a_1* X *w* and from the *βtub85D_-cas9$^{20F}$/+* daughter control. The primers RpS6_1F and RpS6_1R were used to amplify a 234 bp DNA fragment spanning the *RpS6_2* target site. The amplicons were purified with NEB Monarch PCR & DNA Cleanup Kit and quantified with Nanodrop. 200 ng were checked on a gel and 500 ng were sent to GENEWIZ to be sequenced with NGS-based amplicon sequencing. We ran CRISPResso [41] software on raw sequencing data to detect mutations at the target site using parameter -q 30, setting the minimum average read quality score (phred33) to 30.

### Viability studies

To identify the developmental stage at which the progeny from *Muc14a_6/βtub85D_cas9* and *RpS6_2/βtub85D_cas9* crossed to *w$^{1118}$* die, we quantified egg hatching, pupae and adult death rates in the F1. To quantify the egg hatching rate, 20–30 heterozygous *Muc14a_6/βtub85D_-cas9* or *RpS6_2/βtub85D_cas9* and 20–30 *w$^{1118}$* virgin females were set up in embryo collection cages with grape juice agar plates and yeast paste. Two different embryo collection cages with *w$^{1118}$* and *βtub85D_*cas9/+ males crossed to *w$^{1118}$* females served as a comparison control. Between four and six collections of 70–200 embryos each, were performed for each genotype.

Every embryo collection was transferred in a separate fly vial and followed for over 36 h to count the number of embryos that did not hatch, the number of pupae, and female and male adults. Percent survival to each stage was calculated as the ratio between the number of individuals counted and the number of individuals expected for each genotype. The data for the two experiments and for each of the crosses are shown in Table Fig 2C and Table Fig 4C in supplementary Dataset 1.

## Supporting information

**S1 Fig. Evaluation of Cas9 activity when expressed from *βtub85D-cas9* and *βtub85D_cpf1* transgenes integrated at various genomic locations (chromosome indicated in parentheses) using ϕC31 (*attP*) or *piggyBac* mediated transformation.** Flies carrying *βtub85D-cas9* expressed from *attP* docking #15–1 on the second and #10–1 and #4 on the third chromosomes and from piggyBac mediated random integrations #20D on the X, #20 G on the second and #20F and #20C on the third chromosomes were crossed to lines transgenic for *w_ex3-2* gRNA that targets the *white* gene on the X-chromosome. *βtub85D-cas9/w_ex3-2* F1 males with red eyes were then crossed to *white* mutant females, and the female progeny scored for white eyes. Experiments combining two *βtub85D-cas9* transgenes with the *w_ex3-2* gRNA and *βtub85D-cas9* with *w_ex3-2* expressed as part of a gRNA multiplex array were also performed. Similarly, *βtub85D_cpf1/w_ex3-1* red eye males were crossed to *white* females, and the female progeny scored for white eyes. Along with the *βtub85D* promoter, the *nanos-cas9* transgene efficiency was tested with the same *w_ex3-2* gRNA. gRNAs used for each experiment are shown below the graph. n is the number of individuals (males + females) in the F1 progeny.
(TIF)

**S2 Fig. Candidate X-kmer abundance and chromosomal distribution. (A)** Coverage of all candidate X-kmers (grey dots) including those X-kmers chosen for experimental evaluation (colored dots) within the Illumina and PacBio whole-genome sequencing read datasets as predicted by Redkmer. **(B)** Genomic distribution of all candidate X-kmers (grey dots) and those selected X-kmers for experimental testing (colored dots) on *D. melanogaster* chromosome arms based on perfect complementarity. The X-axis indicates the Mbp position of matches on each chromosomal arm, including the 4[th] chromosome, unmapped contigs and heterochromatin (_h).
(TIF)

**S3 Fig. Kmer sub-selection applied to Redkmer output.** Criteria for candidate kmer selection (X-axes) are shown for each of the eight selected X-kmers for X-shredding (colored dots) and for the remaining candidate X-kmers (grey dots). Red vertical lines highlight the minimum cutoff values imposed for the final target site selection. Density plots of each criteria are also shown for the entire Redkmer candidate X-kmer output. The part of the kmer sequence that represents the target sites of experimental gRNAs is indicated (underline).
(TIF)

**S1 Table. Reported in the table are the numbers and percentages of female progenies from different genetic crosses using one or two *cas9* or the *cpf1* endonucleases driven by either the *βtub85D* or the *nanos* promoters, to assess the efficiency of cleavage in the *white* gene.** All crosses are between males bearing the endonuclease and the sgRNA, *w_ex3-2* and *w_ex3-1* for *cas9* and *cpf1* respectively, and females of the *white* genotype. Different *βtub85D-cas9* lines, generated by *attP* or *piggyBac* integrations, are compared. The *Array1_1* is a multiplex of 4 different gRNAs (*Muc14a_4*, *hydra_1*, *esi-2.1_1* and *w_ex3-2*).
(XLSX)

**S2 Table. Predicted perfect blastn hits of kmer target sequences for X-shredding within the *Drosophila melanogaster* heterochromatin-enriched genome assembly [13].**
(XLSX)

**S3 Table. Reported in the table are the numbers and percentages of male and female progenies from genetic crosses between of *βtub85D-cas9/gRNA* and wild-type *w* females to assess the sex-ratio.** sgRNA lines are complementary to single (*Muc14a_1*, *Muc14a_2*, *RpS6_1*, *RpS6_2*, *RpS5a_1* and *RpS5a_2*) or multiple hits on the X chromosome. Progeny of *βtub85D-cas9/+* males crossed to wild type females (no gRNA) or from the reverse cross (*βtub85D-cas9/gRNA* females crossed to wild type males = female control) were used as control. Crosses were set as pools of males and females or as multiple male single crosses for a minimum of ten independent single crosses.
(XLSX)

**S4 Table. Analysis of combinations of gRNAs and Cas9 sources for X-shredding. Reported in the table are the numbers and percentages of males and females in the progeny from genetic crosses between of *βtub85D-cas9/gRNA* and wild-type *w* females.** Average frequencies (%) and standard deviations (STDEV) are shown for each parental genotype with a single copy of *βtub85D-cas9* transgene (1X), two copies of *βtub85D-cas9* transgene (2X) or one copy of *nos-cas9* (grey bars). All lines were crossed to wild type *w* individuals. The reciprocal cross (female ctrl) or heterozygote *βtub85D-cas9/+* or *nos-cas9/+* without gRNA (no gRNA) were used as control. Crosses were set as pools of males and females or as multiple male single crosses for a minimum of ten independent crosses.
(XLSX)

**S5 Table. Survival analysis of the F1 progeny of *Muc14a_6/βtub85D-cas9* males crossed to *w* females compared to *w* and *βtub85D-cas9/+* control males crossed to *w* females.** F1 embryos were collected six, five and four times from cages containing males of the genotype *Muc14a_6/βtub85D-cas9*, *+/βtub85D-cas9* or *white* respectively, crossed to *white* females. The table shows the numbers of embryos, hatched embryos, pupae, females, males and total adults for each collection. The table also shows the percentages of pupae on collected embryos, adults on pupae, and frequencies of males and females on adults.
(XLSX)

**S6 Table. Influence of DNA repair and fusome integrity on X-shredding.** The table shows male and female numbers, frequencies (%) and standard deviation (STDEV) in the F1 of *Muc14a_6/βtub85D-cas9* males with *DNAlig4^{57}*, *spn-A^{057}* or *spn-A^{093}* mutations crossed to *white* females. Also shown are the F1 female and male frequencies for the cross of *UAS_α-spectrinRNAi*; *nos-GAL4* (yellow) knockdown mutants. At least ten independent single crosses per experiment were performed and each cross was between one male and three females.
(XLSX)

**S7 Table. Efficiency of single gRNAs or combinations of gRNAs for X-poisoning.** The table shows the numbers of males, females and percentage of males in the progeny from *βtub85D-cas9* males combined with *gRNA* lines and crossed to wild type *white* females. Individual gRNAs, *RpS5a_1* or *RpS5a _2*, and *RpS6_1* or *RpS6_2* or double gRNA arrays co-expressing *RpS6_2 + RpS5a_1* or *RpS6_2 + RpS6_1* as well as a combination of both *RpS6_2* and *Muc14a_6* transgenes are shown. Control experiments consist of *βtub85D-cas9/+* (no gRNA) males crossed to *white* females. The table also shows the results from the cross between *nos_-cas9/RpS6_2* males and *white* females. The last two columns in the table show the average

number of F1 individuals per single cross and the fraction of experimental versus control (set as 1 for control) averages. Each cross was between one male and three females and at least ten independent single crosses per experiment were performed.
(XLSX)

**S8 Table. Survival analysis of the F1 progeny of *RpS6_2/βtub85D-cas9* males crossed to *w* females compared to *w* and *βtub85D-cas9*/+ control males crossed to *w* females.** F1 embryos were collected six, and five times from cages containing males of the genotype *RpS6_2/βtub85D-cas9*, +/*βtub85D-cas9* or *white* respectively, crossed to *white* females. The table shows the numbers of embryos, hatched embryos, pupae, females, males and total adults for each collection. The table also shows the percentages of pupae on collected embryos, adults on pupae, and frequencies of males and females on adults.
(XLSX)

**S9 Table. Primers used in this study.** The table shows the primers name, description of where they were used and the corresponding nucleotide sequences. The gRNA sequences are underlined.
(XLSX)

## Acknowledgments

The authors would like to thank Steve Russell and Adam Phillippy for helpful discussions and for providing error-corrected whole genome sequence data. We would like to thank Alex Nash, Paolo Capriotti and Rita Colonna.

## Author Contributions

**Conceptualization:** Barbara Fasulo, Philippos Aris Papathanos, Nikolai Windbichler.

**Data curation:** Barbara Fasulo, Angela Meccariello, Maya Morgan, Carl Borufka.

**Formal analysis:** Barbara Fasulo, Philippos Aris Papathanos.

**Funding acquisition:** Nikolai Windbichler.

**Investigation:** Nikolai Windbichler.

**Methodology:** Barbara Fasulo.

**Project administration:** Nikolai Windbichler.

**Resources:** Angela Meccariello.

**Supervision:** Nikolai Windbichler.

**Validation:** Barbara Fasulo, Philippos Aris Papathanos.

**Visualization:** Barbara Fasulo, Philippos Aris Papathanos.

**Writing – original draft:** Nikolai Windbichler.

**Writing – review & editing:** Barbara Fasulo, Philippos Aris Papathanos, Nikolai Windbichler.

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
