## [Decision Letter · Decision Letter 0]

29 Dec 2019

Dear Dr Windbichler,

Thank you very much for submitting your Research Article entitled 'A fly model establishes distinct mechanisms for synthetic CRISPR/Cas9 sex distorters' to PLOS Genetics. Your manuscript was fully evaluated at the editorial level and by independent peer reviewers. The reviewers appreciated the attention to an important topic but identified some aspects of the manuscript that should be improved.

We therefore ask you to modify the manuscript according to the review recommendations before we can consider your manuscript for acceptance. Your revisions should address the specific points made by each reviewer.

[LINK]

Yours sincerely,

Harmit S. Malik

Associate Editor

PLOS Genetics

Gregory Barsh

Editor-in-Chief

PLOS Genetics

Reviewer's Responses to Questions

**Comments to the Authors:**

Reviewer #1: In this study, the authors investigated mechanisms of X-shredders and “X-meddlers” in Drosophila melanogaster. The authors used their previously develop redKmer software to identify suitable repeats on the X-chromosome for shredding. Using autosomal split-Cas9 and gRNA lines, they found that only one gRNA was effective at shredding, and only Cas9 with the beta tubulin promoter, not nanos, despite the latter having overall more cleavage activity. This was likely due to the difference in expression timing between these promoters. The “X-meddler” involved targeting a haplolethal gene on the X-chromosome and represents the first experimental demonstration of a genetic control system based on this strategy (though the final proposed system would need to be on the Y chromosome). This strategy was also successful in biasing the sex ratio toward males.

This manuscript represents a solid step forward in the field and should be published in PLOS Genetics. Here’s some comments on how the authors could potentially improve their manuscript (in roughly the order they appear in the text):

1. The abstract is somewhat light on actual results. Maybe add 2-3 lines regarding the conclusions?

2. In the introduction, the authors say, “This is because the fecundity of females, the sex with a lower rate of gamete production, generally determines the size of a population.” While playing a bit part, I’m not sure if this is technically accurate, particularly given the high density-dependent mortality in mosquito larvae. I’d suggested softening the wording of this phrase.

3. At the end of the first paragraph of the introduction, the authors may want to clarify that the experimental systems have the X-shredder allele on the autosome, so while a big step towards Driving Y elements, they don’t have the power of the Driving Y/X-shredder described in the middle of the paragraph. Additionally, only one of these experimental systems was used in a cage study.

4. It would be nice to have a good term for the X-linked haplolethal targeting strategy, but “X-meddling” is not very informative with regard to the goals of the meddling. I’d encourage the authors to reconsider the name. Maybe “X-poisoning” would be good? Perhaps “X-disrupting” if you want to be less flamboyant?

5. a. For the second paragraph in the results, it may be helpful to mention the specific range of the rate of white-eye offspring for the βtub85D promoter, to make it clear why nanos works better here.

b. Also “LbCfp1” should be “LbCpf1” in one spot. In general, be careful with your cpf1/Cas12a/LbCpf1 terminology.

c. Also, this giant paragraph should probably be broken into smaller more focused paragraphs based on the white results, target determination, and results of the new constructs. I’d suggest adding subsection headings to the results as well.

d. “the RpS5a (McKim, Dahmus, and Hawley 1996), RpS6 (Stewart and Denell 1993) ribosomal protein genes” should be “the RpS5a (McKim, Dahmus, and Hawley 1996) and RpS6 (Stewart and Denell 1993) ribosomal protein genes”.

e. Was the ability of Muc14a_1 and Muc14a_2 to disrupt the target confirmed, as in figure 2A?

6. a. In the third paragraph of results, the authors use the phrase “the ubiquitous Pol III promoter of the Drosophila U6 snRNA gene”, but I think they mean “the Pol III promoter of the Drosophila U6 snRNA gene that drives ubiquitous expression”.

b. The author write, “We also observed that the w_ex3_2 gRNA targeting white, when encoded

as the ultimate of four gRNAs within the array, showed a dramatically reduced level of activity (Figure 2B).” This should be Figure S1. This is likely due to saturation of the Cas9 activity by the gRNAs (this study covers multiplex gRNAs and Cas0 saturation and should be published in Science Advances soon: https://www.biorxiv.org/content/10.1101/679902v1), likely combined with reduced expression of gRNAs in time for cutting by last gRNA due to the tRNA system.

c. Could the term “meiotic activity” be clarified, especially with respect to the usual expression pattern of the two promoters that are compared in the manuscript? For example, is it after meiosis I, or during all of meiosis?

7. a. Statistics are missing on Figures 2C and 4C for the % males and % females (it’s pretty clear, but probably good to add for completeness).

b. Also, it may be better to eliminate the “%” from the category labels. It’s already on the vertical axis, and it could create some confusion in some cases if people go through the figures too quickly (eg, “why don’t the 40% males and 5% females add up to 100% in the brown part of Figure 4C).

c. Also, the legend in 2C has “blue grey white”, but the figure itself has “white grey blue”.

d. There is some redundancy in this figure that could potentially distort useful information. Instead of the current form, it might be better to use egg to pupae survival rates and pupae to adult survival rates instead of absolute % of eggs, and then just have a single column for sex ratio. The sample size could go into a supplement, clearing up the figure quite a bit and putting more focus on what is happening at each stage, rather than having to keep overall trends in mind.

e. Any idea why the βtub85D-Cas9 line had lower hatch rates, especially in 2C?

8. For Figure 4D, only the “A” in “ACC” and the “AT” in “ATG” are highlighted green. However, perhaps all three nucleotides should be highlighted in both cases, representing insertions where some of the nucleotides just happen to be similar to other parts of the sequence?

9. Were the RpS5a and RpS6 targets tried with the nanos-Cas9 line? Since nanos seems to have higher activity, this might substantially increase efficiency. It would be cool to see a little bit of data for this and might be easy to do, but I don’t consider it necessary if the authors don’t have time before submitting their revision.

10. I may have missed this, but could be better gRNAs simply have intrinsically more activity, rather than needed to invoke sequence microenvironments or other explanations? If this is true, it could possibly go against the conclusion that a single gRNA is best. Perhaps a cluster of different gRNAs that each are very active could still be superior, but if a superior gRNA is put with a cluster of inferior ones, it gets too “diluted” and total cleavage is reduced.

11. It looks like the repeats fall within a narrow range, rather than scattered throughout the X-chromosome. Is this different than the Anopheles experiments? It might be worth commenting that targeting a narrow chromosomal window may actually improve the chance for successful repair (since the little pieces that form won’t matter - only the big pieces need to be rejoined. If targets were scattered throughout, then each piece may need to be properly repaired, or at least lots of them). This may not have been seen when multiplexing gRNAs in the current study since only a limited number of gRNAs seemed to be highly active.

12. Comma after “Chromosome-wide, distributed repeats represent an alternative set of targets”.

13. “Alternatively, the observed differences could also partly relate to the predominance of homologous over non-homologous DNA repair pathways acting with varying stringency during the early (including stem cells) and late stages of spermatogenesis, respectively (Chan et al. 2011).” Why would homology-directed repair impact X-shredding when no template for repair is available in males? Perhaps a simple explanation of nanos vs βtub85D X-shredding efficiency is that despite the greater overall activity of nanos, repair is highly efficient in pre/early meiosis and less so in late meiosis when βtub85D has higher activity?

14. “The fly model of X-shredding we have established will allow to tackle these questions experimentally” should instead be something like this: “The fly model of X-shredding we have established will allow these questions to be tackled experimentally”.

15. Maybe comment that X-meddling may be easier to perform from the Y chromosome due to the need for reduced Cas9 activity (just one or a few cuts needed vs. potentially dozens or hundreds). Also, conserved sites wouldn’t necessarily be needed if multiplexed gRNAs were used, as long as they all had good activity.

16. Regarding the possibility of dominant negative mutants in the RpS genes, I think this may certainly be the case. However, could another possibility simply be that the gene isn’t completely haplolethal, and that after passing a certain “critical” stage, likely in embryo development, the gene is haplosufficient? This may be a boring explanation in comparison, but it could explain why you detected some frameshift mutations in viable individuals. This could be assessed by crossing these females (presumably before sequencing) and observing if all the male progeny are viable (in which case, likely dominant negative) or if most are still nonviable (then likely not completely haplolethal). This could be left to a future study.

Minor note: For future submissions, I’d encourage the authors to put the figures and their legends “in-line” with the text if the journal allows submissions like this (I received the same advice when submitting a PLOS Genetics article a while ago, and so far, all journals I’ve worked with seem to allow this now). Bigger spaces between paragraphs would also make the manuscript easier to read.

Overall, good job by the authors. This manuscript should fit in well at PLOS Genetics. I am Jackson Champer, and I’m happy to clarify any of the above points if my wording doesn’t make sense (jc3248@cornell.edu).

Reviewer #2: This manuscript explores the general hypothesis that interfering with X chromosomes during spermatogenesis can impact the sex ratio of progeny derived from affected males. Such a mechanisms, known as X-shredding, has been show to be successful in the malaria mosquito, Anopheles gambiae, and has been proposed as a method for population suppression. However, aside from rare naturally occurring examples of biased sex chromosome transmission, little has been done in Drosophila to look at such systems - given the recent rise in the population of D. suzukii and other dipteran pests, a better understanding of potential genetic control strategies is welcome.

Here the work describes the application of two CRISPR-Cas9 based approaches; X-shredding, where X-specific repeats are targeted for cleavage in the male germline, and X-meddling, a proposed method for sex-biasing populations by targeting halo-insufficient loci such that the homogametic progeny die. Overall, the experiments are well designed and demonstrate that both methods can be applied in Drosophila, suggesting a broader applicability of the approaches, at least in diptera. Key new insights are the use of computational approaches to identify X-specific repeated DNA sequences and showing that at least in one situation targeting such a sequence can bias the sex ratio. Second, the experimental verification of the hypothesis of Burt and Deredec that targeting an X-linked haploinsufficient locus in the male germline could reduce the viability of XX progeny.

While overall the work is interesting and supports the further research into these proposed control mechanisms in a tractable system, I feel the current manuscript leaves much hanging. In particular, there is minimal discussion of, or investigation into, why only one of the Muc14a gRNAs is sucessfull in vivo; aspects of the mutation spectrum generated with the early nos-Cas9 driver versus the later B-tub-Cas9 are poorly explained, in particular the expected outcomes when repair is via HDR or NHEJ is poorly explored. I note that the use of the attached X stock to isolate cut chromosomes in the male is elegant, and the assays in repair pathway mutants are good. I found the discussion around Fig 4 patchy and lacking firm conclusions, while targeting two X-linked ribosomal protein genes did, as predicted, induce male biased progeny, the mechanism for this remains unclear and I would have expected more than just a suggestion it is a dominant effect.

In places the manuscript appears hastily prepared and lacks clarity, particularly in the early sections describing the generation and assay of the Cas9 lines.

Minor comments:

Pg3 & Fig S1- lack of clarity on the Cas9 lines used in the white assays.

FigS1 Legend is wrong #20G on 2nd and #20F and #20G on 3rd

Pg3 & FigS2 & S3 - Lacks clarity on Redkmer selection - matches to Unmapped and unmapped extra reads should be properly explained.

Pg4 & Fig1 Results with esi2 seems odd, generating female bias, and is not commented upon

Pg4 why is there is no female control for Muc14a_6

Pg5 data on w with gRNA array is in fig S1not Fig 2

Pg5 Comment on the lethality associated with the Cas9 alone

Pg5 & Fig2C, male bias looks much less than in 2B?

Pg6 & Fig 3B - why less replicates with Control & nos-Cas9

Pg6 & Fig 3C - helpful if changes from wild type allele were highlighted

Reviewer #3: In this paper, Fasulo and colleagues create experimental synthetic sex ratio distorters in Drosophila melanogaster. “X-shredding” and “X-meddling” are two distinct strategies that were confounded in previous experiments. The experiments in this paper demonstrate that “X-shredding” can work outside of mosquito and therefore could be promising for uses beyond malaria control.

I think that this paper is broadly interesting. I like that the authors report on differences in timing (nos versus Btub85D) and the induced mutations, as these experiments give some insight into the underlying repair processes. The methodology in the paper is straightforward. Below I make some suggestions that I hope will improve the manuscript.

Major comments:

I do not see any tables except for Table S1. I only see reference to Tables 1 and S1, 3 and 5 in the manuscript.

Figure 3 legend: “percentage of all mapping reads” is confusing. Do you mean the percentage of all mapped reads? These data should appear somewhere. In panel B, why do you think that the control only has 90%? Why would nos-cas9 have fewer unique alleles lacking full gRNA target site than the control?

Figures 2C and 4C: I think that you need a better description of the categories (e.g. % male and % female) in the legend. The % hatching is lower for Btub85-cas9/+ (Fig 2C). Is this chromosome different from the Btub85-cas920F used in X-meddler experiments?

Figure 3A: I like the schematic showing the PCR and amplicon sequencing but I had a few questions. Are the thin arrows the primers? And does the 153 bp region correspond to the repeat unit? What are the grey bars flanking the blue bar?

Ideally, the repeats targeted for X shredding would be X-specific for future applications, to reduce the chance of off target effects that may be transmitted. Although your sex ratio data show that the X chromosome is likely targeted, it would be good to give the reader an idea of how enriched these repeats are on the X chromosome. In Figure S2B, you show that the Muc14a target sites are frequent on ‘Unmapped_Extra’ and ‘Unmapped’ and so their locations are unknown. It may not be possible to figure out where all of these repeats are with current technology, but there are at least three things that you could do to add support for the majority of these repeats being X-linked: 1) FISH on polytene and mitotic chromosomes, 2) map to an assembly that has better representation of heterochromatin (https://doi.org/10.1534/genetics.118.301765), 3) plot Illumina depth in male vs female.

The proof-of-principle experiments described in this paper are a necessary and important step towards developing tools intended for vector control. However, I think that it is appropriate to add a sentence or two that mentions some of the caveats for using an X shredder designed to target X-linked repeats. For example, targeting an X-enriched repeat that is dispersed in the genome, or even dispersed on the chromosome, may cause genome rearrangements of unknown consequence that can be transmitted. I think that you need to know more about the targets and their distribution in the genome (and repair processes involving repeats).

Minor:

Figure 2: The figure was not completely clear to me (“a typical region of the Muc14a gene…”) until I saw Figure 3. It might help to add more labels, or more description to the legend. Alternatively, combining with the schematic in Figure 3 could help.

Page 3: About the Drosophila X-linked repeats, it’s not clear what you are comparing to when you say that they are less abundant.

Page 4: It would help to describe the target repeats in a bit more detail here (e.g. repeat unit, distribution, composition).

Pages 4 and 5: Cytology on sperm (e.g. FISH with X-satellite probe) may help you determine if these sperm are lost.

Pg 5, line 4 “the ultimate of four gRNAs” is unclear.

Page 6 lines 1-4: The wording here is hard to follow.

**Have all data underlying the figures and results presented in the manuscript been provided?**

Reviewer #1: Yes

Reviewer #2: Yes

Reviewer #3: No: I could not find any tables except Table S1

PLOS authors have the option to publish the peer review history of their article (what does this mean?). If published, this will include your full peer review and any attached files.

Reviewer #1: No

Reviewer #2: No

Reviewer #3: No

---

## [Editor Report · Decision Letter 1]

3 Feb 2020

Dear Dr Windbichler,

We are pleased to inform you that your manuscript entitled "A fly model establishes distinct mechanisms for synthetic CRISPR/Cas9 sex distorters" has been editorially accepted for publication in PLOS Genetics. Congratulations!

Yours sincerely,

Harmit S. Malik

Associate Editor

PLOS Genetics

Gregory Barsh

Editor-in-Chief

PLOS Genetics

**Data Deposition**

http://datadryad.org/submit?journalID=pgenetics&manu=PGENETICS-D-19-01926R1

**Press Queries**

---

## [Editor Report · Acceptance letter]

6 Mar 2020

PGENETICS-D-19-01926R1 

A fly model establishes distinct mechanisms for synthetic CRISPR/Cas9 sex distorters 

Dear Dr Windbichler, 

We are pleased to inform you that your manuscript entitled "A fly model establishes distinct mechanisms for synthetic CRISPR/Cas9 sex distorters" has been formally accepted for publication in PLOS Genetics! Your manuscript is now with our production department and you will be notified of the publication date in due course.

With kind regards,

Kaitlin Butler

PLOS Genetics

On behalf of:
